# Evaluation of a New Monoclonal Chemiluminescent Immunoassay Stool Antigen Test for the Diagnosis of *Helicobacter pylori* Infection: A Spanish Multicentre Study

**DOI:** 10.3390/jcm11175077

**Published:** 2022-08-29

**Authors:** Elena Resina, María G. Donday, Samuel J. Martínez-Domínguez, Emilio José Laserna-Mendieta, Ángel Lanas, Alfredo J. Lucendo, Marta Sánchez-Luengo, Noelia Alcaide, Luis Fernández-Salazar, Luisa De La Peña-Negro, Luis Bujanda, Marta Gómez-Ruiz de Arbulo, Javier Alcedo, Ángeles Pérez-Aísa, Raúl Rodríguez, Sandra Hermida, Yanire Brenes, Olga P. Nyssen, Javier P. Gisbert

**Affiliations:** 1Gastroenterology Unit, Hospital Universitario de La Princesa, Instituto de Investigación Sanitaria Princesa (IIS-Princesa), Universidad Autónoma de Madrid (UAM), and Centro de Investigación Biomédica en Red de Enfermedades Hepáticas y Digestivas (CIBERehd), 28006 Madrid, Spain; 2Hospital Clínico Universitario Lozano Blesa de Zaragoza, Instituto de Investigación Sanitaria de Aragón (IIS Aragón) and Universidad de Zaragoza, 50009 Zaragoza, Spain; 3Laboratory Medicine Department, Hospital Universitario de La Princesa, 28006 Madrid, Spain; 4Department of Gastroenterology, Hospital General de Tomelloso, 13700 Tomelloso, Spain; 5Instituto de Investigación Sanitaria La Princesa (IIS-Princesa), 28006 Madrid, Spain; 6Instituto de Investigación Sanitaria de Castilla-La Mancha (IDISCAM), 28006 Madrid, Spain; 7Centro de Investigación Biomédica en Red Enfermedades Hepáticas y Digestivas (CIBERehd), 28006 Madrid, Spain; 8Gastroenterology Unit, Hospital Clínico Universitario de Valladolid, Gerencia Regional de Salud (SACYL) and Universidad de Valladolid, 47003 Valladolid, Spain; 9Gastroenterology Unit, Hospital de Viladecans, 08840 Barcelona, Spain; 10Gastroenterology Unit, Hospital Donostia/Instituto Biodonostia, Universidad del País Vasco (UPV/EHU), Centro de Investigación Biomédica en Red de Enfermedades Hepáticas y Digestivas (CIBERehd), 20014 San Sebastián, Spain; 11Microbiology Department, Hospital Donostia/Instituto Biodonostia, Centro de Investigación Biomédica en Red de Enfermedades Respiratorias (CIBERES), 20014 San Sebastián, Spain; 12Gastroenterology Unit, Hospital Universitario Miguel Servet, 50009 Zaragoza, Spain; 13Gastroenterology Unit, Agencia Sanitaria Costa del Sol, 29651 Marbella, Spain; 14Hospital General Universitario de Castellón, 20014 Castellón de la Plana, Spain

**Keywords:** diagnosis, *Helicobacter pylori*, stool antigen test, urea breath test

## Abstract

The stool antigen test (SAT) represents an attractive alternative for detection of *Helicobacter pylori*. The aim of this study was to assess the accuracy of a new SAT, the automated LIAISON^®^ Meridian *H. pylori* SA based on monoclonal antibodies, compared to the defined gold standard ^13^C-urea breath test (UBT). This prospective multicentre study (nine Spanish centres) enrolled patients ≥18 years of age with clinical indication to perform UBT for the initial diagnosis and for confirmation of bacterial eradication. Two UBT methods were used: mass spectrometry (MS) including citric acid (CA) or infrared spectrophotometry (IRS) without CA. Overall, 307 patients (145 naïve, 162 with confirmation of eradication) were analysed. Using recommended cut-off values (negative SAT < 0.90, positive ≥ 1.10) the sensitivity, specificity, positive predictive value, negative predictive value and accuracy were 67%, 97%, 86%, 92% and 91%, respectively, obtaining an area under the receiver operating characteristic (ROC) curve (AUC) of 0.85. Twenty-eight patients, including seven false positives and 21 false negatives, presented a discordant result between SAT and UBT. Among the 21 false negatives, four of six tested with MS and 11 of 15 tested with IRS presented a borderline UBT delta value. In 25 discordant samples, PCR targeting *H. pylori* DNA was performed to re-assess positivity and SAT accuracy was re-analysed: sensitivity, specificity, positive predictive value, negative predictive value, accuracy and AUC were 94%, 97%, 86%, 99%, 97% and 0.96, respectively. The new LIAISON^®^ Meridian *H. pylori* SA SAT showed a good accuracy for diagnosis of *H. pylori* infection.

## 1. Introduction

*Helicobacter pylori* (*H. pylori*) is a highly prevalent worldwide infection and is the main cause not only of gastritis, but also of peptic ulcer and gastric cancer. Therefore, an accurate diagnosis remains crucial [1]. *H. pylori* screening and eradication is indicated in several clinical contexts including gastric cancer precursor lesions, peptic ulcer disease and gastric mucosa-associated lymphoid tissue lymphoma, among others [2,3]. However, more recently, consensus has been reached on eradicating *H. pylori* regardless of the associated clinical condition [4].

Although different methods have been established to detect *H. pylori* infection, identifying *H. pylori*-infected patients is a challenge. *H. pylori* infection diagnostic methods are classically divided into invasive and non-invasive. Invasive methods are based on the detection of the organism in the stomach by means of gastric biopsy samples (histology, rapid urease test (RUT), culture and molecular diagnostic methods), which require an endoscopy. This strategy should be considered in patients with alarm symptoms or in older (i.e., >50–55 years) patients. On the other hand, non-invasive methods do not require an endoscopic examination and are currently the most widely used diagnostic techniques [5,6,7].

Uninvestigated dyspepsia represents one-quarter of primary care referrals to the gastroenterologist in Spain [8]. In the absence of alarm symptoms, and in those younger than 50–55, a “test-and-treat” strategy would be appropriate for diagnosis, preferring non-invasive tests such as urea breath test (UBT) or stool antigen test (SAT) rather than prescribing proton pump inhibitors (PPIs) or endoscopy [9,10,11].

The UBT has been classically considered the preferred non-invasive technique due to its excellent sensitivity and specificity (>95–100%) demonstrated in countless studies and meta-analyses. Moreover, UBT has been the most widely used diagnostic method in recent years, making it an optimal gold standard [12,13]. Although UBT is safe, some disadvantages have been described: point-of-care testing, significant consumption of patient time (30–60 min), fasting requirement, the high cost of qualified personnel or sending the collected samples to an analytical laboratory [14]. In addition, there are physical difficulties for certain populations (e.g., children and disabled) in performing the test [15].

The SAT is a low-cost diagnostic method, without the previously described disadvantages, suitable for at-risk populations [16]. Indeed, some studies have suggested its better cost-effectiveness compared to other techniques [17,18].

Two types of SATs have been described: the enzyme immunoassays, such as enzyme-linked immunosorbent assay (ELISA) or the more recently developed, chemiluminescence-based immunoassays (CLIA), and “rapid” or “in-office” tests (immunochromatography assay or ICA). These methods can be based either on polyclonal or monoclonal antibodies [6,16]. With regard to “rapid” or “in-office” tests, controversial results have been reported [19] and they are not recommended in clinical practice guidelines due to their limited accuracy [20].

The diagnostic performance of different SATs is heterogeneous, although there is no doubt that the use of monoclonal antibodies has resulted in a higher accuracy compared to polyclonal antibodies, both in adults and children [15,19,21].

The high accuracy of SAT has been widely demonstrated in most reviews and meta-analyses, providing a sensitivity above 80–90%. Additionally, a sensitivity of 83% has been reported in a very recent Cochrane meta-analysis, with no discrimination between monoclonal and polyclonal antibodies [12], while other studies evaluating the monoclonal technique showed a higher sensitivity and specificity (96.2% and 94.7%, respectively), in the paediatric population [21].

Another study, also in children, showed a sensitivity and specificity of 97% [15]. A meta-analysis by Gisbert et al. reported sensitivity and specificity data >93% both before and after *H. pylori* treatment [19]. Additionally, a clinical review by the Canadian Drug Agency described a sensitivity and specificity of 90.0–92.4% and 91.0–100%, respectively, with monoclonal SAT [22]. Therefore, SAT is currently recommended by the main international guidelines as a valid and reliable diagnostic method [2,20,23].

As previously mentioned, the CLIA technique for the identification of *H. pylori* in faeces has proven similar accuracy to that reported for both ELISA and the best performing lateral flow immunoassays, with additional advantages [24].

In the current study, we evaluated the accuracy of the LIAISON^®^ Meridian *H. pylori* SA test for the diagnosis and confirmation of eradication of *H. pylori* infection. This new test is a fully automated, time saving, objective and traceable CLIA, which detects the presence of *H. pylori* antigen in human stools using unique monoclonal antibodies.

## 2. Materials and Methods

### 2.1. Patients and Design

This is a prospective, comparative, multicentre study that aimed to evaluate the accuracy of the new LIAISON^®^ Meridian *H. pylori* SA stool antigen test. Nine Spanish centres participated in the study. Consecutive adult patients with indication of *H. pylori* infection primary diagnosis or confirmation of the bacterial eradication according to standard clinical practice were enrolled between November 2019 and May 2021. Patients were included for both pre- and post-treatment tests, and the signing of an informed consent for each test was required.

Inclusion criteria were indication to perform an *H. pylori* infection diagnosis; capacity and willingness to give written informed consent; prior prescribed ^13^C-UBT; and, for confirmation of eradication (if the patient was included for post-eradication treatment diagnosis), performance of the test at least four weeks after the treatment was discontinued. Exclusion criteria were age below 18 years; advanced chronic disease that would not allow the patient to complete follow-up or attend follow-up visits; previous gastric surgery; alcohol or drug abuse; antibiotic or bismuth salts consumption four weeks prior to testing; and PPI intake two weeks prior to testing.

The study protocol was approved by the Ethics Committee of all participant hospitals.

### 2.2. Stool Antigen Test

Patients were instructed to obtain a faecal sample within 24–36 h either before or after the UBT was performed.

*H. pylori* was detected using a new automated LIAISON^®^ Meridian *H. pylori* SA assay (REF 318200, DiaSorin, Stillwater, MN, USA) (Appendix A). The test is a CLIA in sandwich format that uses novel monoclonal antibodies for capture and detection of *H. pylori* stool antigen. Testing was performed following the manufacturer’s instructions at the laboratory of Hospital de La Princesa (Madrid, Spain). Specimens had to be stored at least at –20 °C.

The process was performed automatically by the LIAISON^®^ XL analyser (DiaSorin SpA, Saluggia, Italy). Two hundred microlitres of the diluted sample (mixture of sample diluent and stool) was incubated with paramagnetic particles coated with capture antibodies. Isoluminol conjugated antibodies for *H. pylori* antigen were subsequently added and incubated and the unbound material was washed. Then the flash chemiluminescent reaction was initiated and chemiluminescent light was measured by a photomultiplier. Relative light units (RLU) were recorded. RLU were proportional to the concentration of the *H. pylori* stool antigen present.

Specimens were classified as negative, equivocal or positive based on their index (<0.9, 0.9–1.1 and >1.1, respectively). The trained operator was unaware of the results of the reference tests.

Equivocal LIAISON^®^ results (CLIA values of 0.9–1.1) were solved by immediately repeating SAT on another sample from the same specimen using a kit from a different batch. This latter result was the one valid for analysis.

### 2.3. Urea Breath Test

The gold standard was defined by the ^13^C-UBT. Two UBT methods were used according to the available and standard clinical practice from the participating centres. The professionals analysing the UBT test were unaware of the SAT results.

#### 2.3.1. Isotope Ratio Mass Spectrometry (IRMS)

The commercial test TAU-KIT^®^ (Isomed, S.L., Madrid, Spain) was used. The patient had to fast at least six hours before the test was performed. Initially, a citric acid solution (4.2 g, Citral pylori^®^) dissolved in 200 mL of water was administered. After 10 min, samples were collected to determine the reference value of the test. The urea solution was prepared by dissolving a 100 mg tablet of this substrate in 125 mL of water. After 30 min following the administration of the urea solution, samples of exhaled breath were again collected. Samples were classified as either negative, unclear or positive based on delta values (<4‰, 4–5‰ and 5‰, respectively).

Patients with unclear UBT results (delta values between 4‰ and 5‰ measured by mass spectrometer) were asked to repeat both the UBT and the stool sample at least four weeks after testing. These latter results were used for the analysis.

#### 2.3.2. Non-Dispersive Isotope-Selective Infrared Spectrometry (NDIRS)

The UBT was performed using UBTest 100 mg (Ferrer Internacional, Barcelona, Spain). Determinations were performed in accordance with the manufacturer’s specifications. The patient had to fast at least eight hours before the test performance. A basal breath sample was collected by blowing into a specially designed bag. After this, patient swallowed a pill of 100 mg of ^13^C-labelled urea in 100 mL of water, and 20 min later filled a second breath bag. Samples were immediately processed by NDIRS (POCone^TM^ or POConePlus, Infrared Spectrophotometer, Otsuka Pharmaceutical, Tokyo, Japan). In accordance with the manufacturer’s specifications, an increase in the proportion ^13^C/^12^C (Δ^13^CO_2_ (‰)) of 2.5‰ or more after urea intake was considered as indicative of *H. pylori* infection. No unclear values are described by the manufacturer.

### 2.4. H. pylori Detection by PCR

Faecal samples with discrepant results between UBT and SAT data were subjected to two different polymerase chain reactions (PCR) in an external laboratory for confirmation of *H. pylori* presence. Samples were transported at a temperature of at least −20 °C. First, DNA was extracted in an automated extractor device (KingFisher, Thermo Scientific, Waltham, MA, USA) employing magnetic particles (MagMax CORE Nucleic Acid Purification Kit, Applied Biosystems, Foster City, CA, USA) and following the instructions from the manufacturers. PCR amplification was performed in 100 ng of DNA (final volume 50 µL) by using primer oligonucleotides for the glmM gene and the 16S rRNA region of *H. pylori* (SYNLAB Diagnósticos Globales S.A.U., Barcelona, Spain). The threshold detection limit of the aforementioned techniques is 200–300 copies per gram of faeces. For the amplification of the *H. pylori* 16S rRNA region, a nested PCR assay was performed with 30 cycles in each amplification and hybridisation temperatures of 34 °C and 41 °C in the first and second amplifications, respectively. For the amplification of the glmM gene, a PCR assay was performed with 40 cycles at a hybridisation temperature of 48 °C. Detection of the specific amplified bands (398 bp for 16S rRNA and 294 bp for glmM gene) was performed in 3% agarose gels. All tests were performed by experienced laboratory professionals, blinded to the *H. pylori* status of the samples and to the results of the other tests. To confirm the presence or absence of *H. pylori* in faecal samples, the results of both PCRs must be positive or negative, respectively.

### 2.5. Statistical Analysis

Quantitative variables are provided as the mean and the standard deviation (SD). Categorical variables are expressed as percentages and 95% confidence intervals (CIs). The statistical significance threshold was set at a *p*-value < 0.05.

Sensitivity, specificity, positive predictive value (PPV) and negative predictive value (NPV) and their 95% CIs were calculated by standard methods, and positive and negative likelihood ratios (LR+ and LR–, respectively), using the UBT results as study gold standard. The study was reported in compliance with the Standards for the Reporting of Diagnostic Accuracy Studies (STARD) recommendations [25].

The sample size was calculated based on the formula by Jones et al. [26]. Two different sample size calculations were performed: one each for the pre-treatment and post-treatment analyses.

Calculations were performed using an expected (desired) sensitivity and specificity of 97%, a precision of 5% and a CI of 95%. Based on published data and previous experience in our country, the expected pre-treatment prevalence was 45% (of dyspeptic patients attending a gastroenterology outpatient clinic) and post-treatment prevalence would range from 20 to 30% depending on the administered treatment (as the effectiveness in clinical practice ranges from 70 to 80%).

Under these conditions, the estimated required sample size was 100 and 166 patients for pre-treatment and post-treatment, respectively. It should be taken into consideration that some doctors are prescribing optimized treatments (four drugs, 14 days), which may reduce the post-treatment prevalence down to 10–15%. In the worst case scenario (with only a 10% post-treatment prevalence), given this sample size, and maintaining a 95% CI and an expected accuracy at 97%, the precision would be 8%.

## 3. Results

A total of 321 patients were screened in nine centres; of these, 14 were excluded, leaving 307 patients for enrolment in the study (Figure 1). The baseline characteristics of the included subjects are shown in Table 1.

A STARD flow diagram of the study is shown in Figure 2, including the overall pool of patients evaluated, the pre-treatment group (i.e., treatment-naïve patients) and the post-treatment group (i.e., those where a confirmation of eradication was requested).

In two patients, delta values were unclear (between 4‰ and 5‰, measured by IRMS). In these two patients, the UBT was repeated, and a new stool sample was collected at least four weeks after the baseline test. Another two faecal samples provided equivocal LIAISON^®^ results (CLIA values of 0.9–1.1). This problem was solved by immediately repeating the SAT on another sample from the same specimen, obtaining valid results.

The LIAISON^®^ Meridian *H. pylori* SA test had an overall sensitivity of 67% (95% CI 55–79%) and a specificity of 97% (95–99%). PPV and NPV were 86% (75–97%) and 92% (88–95%), respectively. The LR+ and LR– were 23 (11–49) and 0.34 (0.24–0.48), respectively. Global accuracy was 91% (88–94%) with an area under the ROC curve (AUC) of 0.85 (0.78–0.92).

Sensitivity, specificity, PPV, NPV, accuracy, LR+, LR–, and AUC were calculated separately for treatment-naïve patients and in the post-treatment group. Results are shown in Table 2 and Table 3, respectively.

In total, 28/307 patients (including seven false positives and 21 false negatives) presented a discordant result between SAT and UBT. Among the 21 false negatives, four of six tested with IRMS had a UBT delta value very close to the cut-off point (values between 5.2 and 5.8 being the cut-off point with IRMS 5‰). In addition, 11/15 tested with NDIRS had a UBT delta value near the cut-off point (with values below 10), and 10/15 had values equal to or lower than 8.5 (the cut-off point with NDIRS 2.5‰).

Twenty-five of these samples were subjected to a confirmatory PCR. In three discordant samples, all of them classified as false negatives, the remaining sample was insufficient for PCR analysis. All stool PCRs were negative for *H. pylori* DNA detection. The accuracy of the LIAISON^®^ Meridian *H. pylori* SA assay was re-analysed, considering these patients as non-infected. Results are shown in Table 4.

Mean UBT value (±SD) in treatment-naïve positive patients was 43.3 ± 28.4 and 18.4 ± 15.2 measured by IRMS and NDIRS, respectively. Mean UBT value (±SD) in post-treatment negative patients was 0.96 ± 0.98 and 0.54 ± 0.52 measured by IRMS and NDIRS, respectively.

## 4. Discussion

The present study evaluated the accuracy of an *H. pylori* diagnostic method, a chemiluminescent immunoassay of the LIAISON^®^ Meridian *H. pylori* SA versus the performance of the ^13^C-UBT, defined as the gold standard.

Although high levels of global accuracy (91%) and specificity (97%) were shown with this novel strategy, the sensitivity was sub-optimal (67%) both in naïve diagnostic testing (74%) and in post-treatment eradication confirmatory testing (55%). Even though the results were lower than expected, they were within the range of variability found in a recent meta-analysis of SATs, which described a high heterogeneity of performance depending on the brand, or even for the same brand in different populations [22]. Several methodological aspects should be taken into consideration when extracting conclusions from our results.

First, the choice of ^13^C-UBT as the unique gold standard method might represent a limitation in the correct classification of patients as positive or negative for *H. pylori* infection. The concern arises because no invasive diagnostic method, such as histological evaluation, RUT or culture was used as the gold standard, neither a combination of two or more diagnostic tests. Therefore, using ^13^C-UBT only as the reference method, misdiagnosis by the UBT may hide the existence of correct stool classifications by SAT.

Second, and regarding the two different UBT technologies used, multiple studies and meta-analyses support the high accuracy of NDIRS and its high correlation with IRMS, with sensitivity and specificity for *H. pylori* infection diagnosis > 90–95% for both techniques [27,28]. Conversely, recent studies [29,30,31] have suggested that NDIRS tests offer a low specificity, between 47 and 88%, and a low PPV (with Otsuka equipment, including the POCone spectrophotometer used in our study), with a relatively high false-positive rate. This could be a consequence of the NDIRS cut-off point set by the manufacturer, as there is evidence of an increase in specificity from 60 to 90% [30], or from 47.1% to 95.7% [31] by raising the currently recommended cut-off point from 2.5‰ to 8.5‰. In fact, other studies [29,31] suggest an even larger “grey or borderline area” of 2–10‰ where the positive UBT results obtained with NDIRS may be less reliable and could possibly be classified as false positive.

Kwon et al., basing their results on a post-treatment group of 223 UBT-positive patients with values ranging from 2.5‰ to 10.0‰, found that 34% were false positives as determined by endoscopic biopsy [31]. Furthermore, there is evidence that when the UBT is positive, delta figures are usually much higher than the cut-off point [32], in line with our study, where results showed mean positive UBT values of 43 and 18 measured by IRMS and NDIRS, respectively.

In this respect, it should be highlighted in our study that the majority of “false negatives” of the SAT (11/15 analysed with NDIRS) were within these borderline values (2.5‰ to 10.0‰). In addition, very recent guidelines recommended the use of IRMS technology as a first choice UBT reference method, leaving NDIRS as an alternative [33]. Furthermore, it is remarkable that citric acid is not included in the NDIRS protocol, in contrast to that of IRMS, as citric acid has proven to make the UBT more robust and is an essential component of the protocol [2,13].

All the above-mentioned considerations might explain the lower sensitivity of the LIAISON^®^ Meridian *H. pylori* SA assay (65%) in the present study when compared to NDIRS, which raised (to 71%) when compared to IRMS. These data suggest a slightly lower PPV and specificity of the NDIRS UBT than would be expected. Moreover, it can be assumed that the higher accuracy of the IRMS UBT is probably due, at least in part, to the use of citric acid in its protocol.

Focusing on the benefits of using SAT as a non-invasive method, several advantages such as rapidity are to be highlighted, since the sample can be delivered with no need to remain in the hospital or medical centre, thereby reducing work absenteeism, tending to simplicity in the procedure and subsequently lowering the costs. In addition, some studies suggested that SAT has the same diagnostic value in patients with distal gastrectomy as in patients without surgery [14,34] and regardless of treatment with PPIs [35]. However, these statements are not widely accepted, and further studies are needed. Hence, we excluded these patients from our protocol. On the other hand, the SAT also has some disadvantages, including that negative SAT results may not indicate the absence of *H. pylori* infection but rather low density of *H. pylori* in the stomach and a low antigen load in the stool [16]. However, this disadvantage is common for most non-invasive methods, for instance, with UBT it has even been suggested to select a lower cut-off value (compared to the pre-treatment setting) in order to maintain the diagnostic accuracy for the monitoring of *H. pylori* eradication [32].

Concerning the SAT methodology, as mentioned in Section 1, SATs can be either based on ELISA, CLIA or ICA [7]. The former has been the most widely studied and recommended by international guidelines due to its high diagnostic accuracy [2]. Laboratory-based CLIAs are widely used in clinical laboratories. This technique has elevated sensitivity due to the broad contact surface area provided by the particles of the reagent for antigen–antibody interaction, as well as the increased intra- and inter-assay accuracy inherent to CLIA versus ELISA methodology. Luminescent transitions of excited molecules or atoms to a state of lower energy are characterised by electromagnetic radiation dissipated as photons in the ultraviolet (UV), visible or near-infrared region. These luminescent reactions are classified according to the energy source involved during the excitation step; thus, most classical light-emission reactions are referred to as bioluminescence (from in vivo systems), chemiluminescence (from a chemical reaction, as is the case of this study protocol), electroluminescence (from electrochemical reaction) and photoluminescence (from UV, visible or near-infrared radiations). Chemiluminescence reactions are generally oxidoreduction processes and the excited compound, which is the reaction product, has a different chemical structure from the initial reactant [36,37]. The technology has the ability to detect minimal quantities of antigen in stool samples (4 ng/mL), and by using a photomultiplier, the luminescence signal can be measured down to a few photons [24]. Additionally, automation enables processing a large number of samples in a short period of time, thereby saving time, providing objective results, minimising mishandling and ensuring proper traceability. Altogether these characteristics warrant validation studies of this technique in our setting.

Technological advances allow the use of techniques such as chemiluminescence for the diagnosis of *H. pylori* infection, and technological developments also enable progress in the treatment or control of infection through the use of sensors capable of detecting anti-biofilm drugs or urease inhibitors [38,39].

The accuracy of the automated LIAISON^®^ Meridian *H. pylori* SA assay has been previously evaluated in a limited number of publications. Opekun et al. [40] included 277 patients and combined different methods including histology, culture and RUT as gold standard, obtaining a high sensitivity and specificity (95.5% and 97.6%, respectively). However, only eight patients were included in the post-treatment group. Ramírez-Lázaro et al. [24] included 252 untreated patients. Compared to the gold standard (concordance of the RUT, histopathology and UBT), the test had a sensitivity of 90.1% and a specificity of 92.4%, with PPV and NPV > 90%. Another evaluation of the test in 103 untreated patients with dyspepsia was performed in Spain [41] using the same gold standard (UBT) as in the present study and reporting results similar to ours, with a sensitivity of 72% and a specificity of 96.2%. However, this latter study was reported only as a conference abstract. Thus, to the best of our knowledge, our study included the largest number of post-treatment patients in which confirmation of *H. pylori* eradication was assessed.

It has been suggested that in the post-treatment setting the SAT may have a suboptimal performance given the low prevalence of infection, with PPV around 50% [42]. In this regard, the present study is in accordance with current literature as the LIAISON^®^ Meridian *H. pylori* SA assay accuracy was inferior in the post-treatment setting (sensitivity 55%, specificity 98%, AUC 0.79) than in naïve patients (sensitivity 74%, specificity 96%, AUC 0.88) while maintaining an acceptable post-treatment PPV of 80% despite a prevalence of *H. pylori* of only 14% (consequence of an eradication rate of 86%). Therefore, despite the fact that meta-analyses claim that *H. pylori* monoclonal SAT can also be used in the post-treatment setting with high sensitivity and specificity [19], it may have worse diagnostic performance than in the naïve setting. Nevertheless, considering the data after performing PCR on stool from discordant samples, excellent accuracy values were also obtained in the post-treatment setting (sensitivity 100%, specificity 98%, AUC 0.94).

The use of molecular techniques, such as stool PCR, is increasing not only for diagnosis but also for non-invasive detection of bacterial antibiotic resistance [43]. The overall sensitivity of stool PCR as a diagnostic test, depending on target gene, ranges between 40 and 100%, being the highest for the 23s RNA and 16s RNA genes, the latter with sensitivity and specificity close to 100% in some studies [44]. The two most consistent meta-analyses, one including twenty-six studies and the other one including seven studies in children, calculated a sensitivity range of 70–80% [15,45].

The genes used in our study, glmM and 16s rRNA, have a sensitivity of 56% and 74%, and a high specificity of 99% and 87%, respectively, with an excellent AUC (≥0.95), placing them as two of the first-choice genes for stool PCR testing [45]. In addition, we used a nested PCR for the amplification of the 16s rRNA gene, a front-line technique due to its higher sensitivity and the ability to amplify the target sequence at a lower concentration, as it involves two rounds of amplification [15,16,45]. Furthermore, combination of several target genes might help to improve the diagnostic performance by reducing the possibility of missed detection [16]. The combination of these two genes in gastric biopsies of 387 patients provided a sensitivity and specificity of 92.9% and 92.4%, respectively, which is superior to histopathology and RUT [46].

Therefore, we re-analysed the data by considering our 25 discordant patients as “true negatives”, with both negative genes in stool samples, especially in view of the borderline positive value of the UBT in most of them. This improved the accuracy of the LIAISON^®^
*H. pylori* SA up to a sensitivity of 94%, a specificity of 97%, and an AUC of 0.96.

Our study has some limitations, such as not having an invasive method or a combination of several diagnostic methods as the gold standard or using two different UBT technologies. On the other hand, our study has several strengths: the inclusion of prospectively evaluated patients, the multicentre study design, its large sample size, the inclusion not only of pre-treatment but also post-treatment patients, and the performance of stool PCR analysis of discordant samples.

## 5. Conclusions

In conclusion, the LIAISON^®^
*H. pylori* SA chemiluminescent diagnostic assay showed a good accuracy for diagnosis of *H. pylori* infection, both pre- and post-treatment.

## Figures and Tables

**Figure 1 jcm-11-05077-f001:**
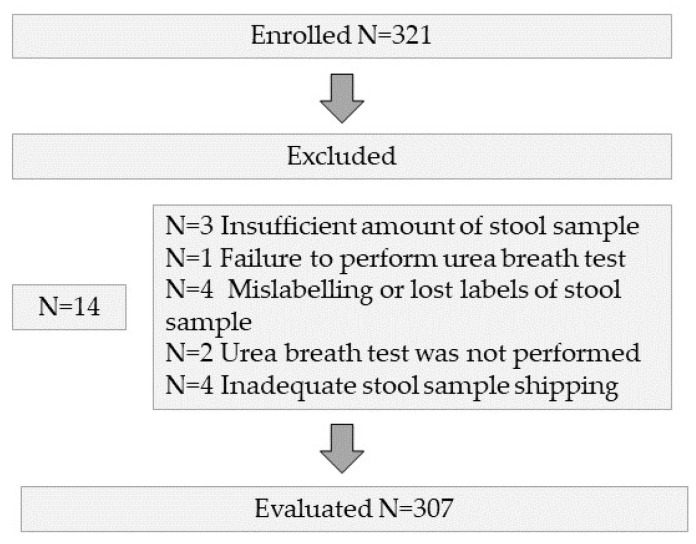
Flow diagram of the enrolled patients.

**Figure 2 jcm-11-05077-f002:**
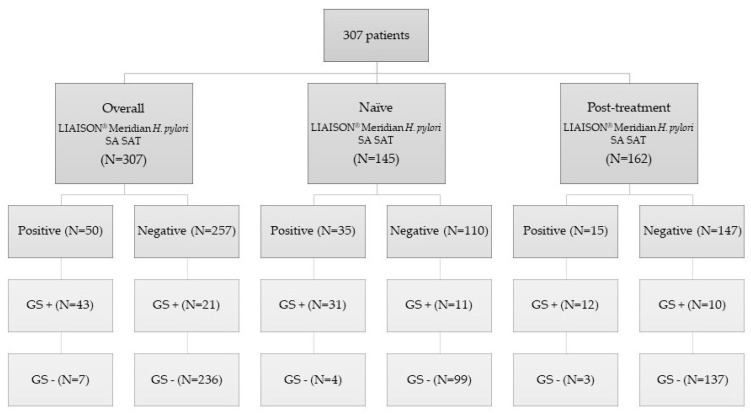
Standards for the Reporting of Diagnostic Accuracy Studies (STARD) flow diagram of the study. N: number of patients included; SAT: stool antigen test; GS+: gold standard positive; GS−: gold standard negative.

**Table 1 jcm-11-05077-t001:** Characteristics of the included patients.

Variables	
Age (mean ± SD)	47.1 ± 14.4
	**N (%)**
Gender (female)	207 (67)
Clinical indication	
○ Pre-treatment/naïve	145 (47)
○ Post-treatment/confirmation	162 (53)
UBT method	
○ IRMS	118 (38)
○ NDIRS	189 (62)
History of peptic ulcer	14 (5)
*H. pylori* infection prevalence (by UBT)	**% (95% CI)**
○ Overall	21 (16–26%)
○ Pre-treatment/naïve	29 (21–37%)
○ Post-treatment/confirmation	14 (8–19%)
	**N = 307**

N: number of patients included; CI: confidence intervals; UBT: urea breath test; IRMS: isotope ratio mass spectrometry; NDIRS: non-dispersive isotope-selective infrared spectrometry.

**Table 2 jcm-11-05077-t002:** Accuracy of the LIAISON^®^ Meridian *H. pylori* SA test in treatment-naïve patients.

Comparison SAT vs. UBT	Sensitivity (95% CI)	Specificity (95% CI)	PPV (95% CI)	NPV (95% CI)	LR+ (95% CI)	LR− (95% CI)	Global Accuracy (95% CI)	AUC (95% CI)
Naïve	74% (59–88)	96% (92–100)	89% (77–100)	90% (84–96)	19 (7–51)	0.27 (0.16–0.45)	90% (84–95)	0.88 (0.80–0.96)
NDIRS naïve	73% (54–92)	93% (86–100)	83% (65–100)	89% (80–97)	11 (4–29)	0.29 (0.15–0.55)	87% (80–95)	0.86 (0.75–0.96)
IRMS naïve	75% (51–99)	100% (99–100)	100% (96–100)	92% (82–100)	-	0.25 (0.11–0.58)	93% (86–100)	0.91 (0.79–1.0)

SAT: stool antigen test; UBT: urea breath test; PPV: positive predictive value; NPV: negative predictive value; LR+: positive likelihood ratio; LR−: negative likelihood ratio; AUC: area under the ROC curve; IRMS: isotope ratio mass spectrometry, cut-off value 5‰; NDIRS: non-dispersive isotope-selective infrared spectrometry, cut-off value 2.5‰.

**Table 3 jcm-11-05077-t003:** Accuracy of the LIAISON^®^ Meridian *H. pylori* SA test in post-treatment patients.

Comparison SAT vs. UBT	Sensitivity (95% CI)	Specificity (95% CI)	PPV (95% CI)	NPV (95% CI)	LR+ (95% CI)	LR− (95% CI)	Global Accuracy (95% CI)	AUC (95% CI)
Post-treatment	55% (31–78)	98% (95–100)	80% (56–100)	93% (89–98)	25 (8–82)	0.46 (0.29–0.73)	92% (87–96)	0.79 (0.65–0.93)
NDIRS post-treatment	53 % (26–80)	99% (96–100)	90% (66–100)	91% (85–98)	46 (6–336)	0.48 (0.29–0.79)	91% (85–97)	0.81 (0.67–0.95)
IRMS post-treatment	60% (7–100)	96% (90–100)	60% (7–100)	96% (90–100)	16 (3–75)	0.42 (0.14–1.22)	93% (86–100)	0.78 (0.44–1.0)

SAT: stool antigen test; UBT: urea breath test; PPV: positive predictive value; NPV: negative predictive value; LR+: positive likelihood ratio; LR−: negative likelihood ratio; AUC: area under the ROC curve; IRMS: isotope ratio mass spectrometry, cut-off value 5‰; NDIRS: non-dispersive isotope-selective infrared spectrometry, cut-off value 2.5‰.

**Table 4 jcm-11-05077-t004:** Accuracy of the LIAISON^®^ Meridian *H. pylori* SA test after performing PCR in stool of discordant samples.

Comparison SAT vs. UBT	Sensitivity (95% CI)	Specificity (95% CI)	PPV (95% CI)	NPV (95% CI)	LR+ (95% CI)	LR− (95% CI)	Global Accuracy (95% CI)	AUC (95% CI)
Overall	94% (85–100)	97% (95–99)	86% (75–97)	99% (97–100)	35 (17–73)	0.07 (0.02–0.20)	97% (95–99)	0.96 (0.91–1.0)
Naïve	91% (80–100)	96% (93–100)	89% (77–100)	97% (94–100)	25 (10–67)	0.09 (0.03–0.27)	95% (91–99)	0.996 (0.99–1.0)
Post-treatment	100% (96–100)	98% (95–100)	80% (56–100)	100% (99–100)	50 (16–153)	0.00	98% (96–100)	0.94 (0.88–1.0)

SAT: stool antigen test; UBT: urea breath test; PPV: positive predictive value; NPV: negative predictive value; LR+: positive likelihood ratio; LR−: negative likelihood ratio; AUC: area under the ROC curve.

## Data Availability

The data presented in this study are available on request from the corresponding author.

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
