# Peer review of "Evaluation of a New Monoclonal Chemiluminescent Immunoassay Stool Antigen Test for the Diagnosis of Helicobacter pylori Infection: A Spanish Multicentre Study"

_jcm, 2022, doi:10.3390/jcm11175077_

Round 1

Reviewer 1 Report

This is a very interesting article on a new diagnostic tool for Hp infection compared to the gold standard. The problem is well presented, the methods and results are clearly explained and the discussion is in depth. The figures and tables are clear and illustrative. To be noted that the statistical analysis is also accurate. The conclusions don´t go further than the analysis of the results can allow. The bibliography is wide enough.

Author Response

We would like to express our sincere gratitude to the reviewers for their valuable work; their annotations have allowed us not only to significantly improve the manuscript but also to reflect on future research. No changes have been made to the manuscript

Reviewer 2 Report

The comments are listed in detail as follows:

1. The first time the word/phrase is introduced, put the abbreviation in brackets after the word/phrase, such as “ROC” in Page 1, line 49. The full name of the “ROC” abbreviation should be given when it first appears.

2. The detection principle of the new monoclonal chemiluminescent immunoas-2 say stool antigen test method should be provided in this manuscript.

3. The detailed testing experimental steps of the new monoclonal chemiluminescent immunoas-2 say stool antigen test method should be supplemented in the section of “2. Materials and Methods”.

4. This paper reports on the evaluation of a new monoclonal chemiluminescent immunoas-2 say stool antigen test for the diagnosis of Helicobacter pylori infection. In a wider perspective, also other approaches, based on immobilization of pathogens or organelles on the sensor surface have been attempted to study the interactions of drugs with sensor components for the purpose of pathogen control or disease curing (Biosensors and Bioelectronics, 2022, 215: 114376; Sensors and Actuators B: Chemical, 2022, 355: 131284). These relevant literature references should be cited for the benefit of general readership.

5. P4, line 196, “2.4 H. pylori detection by PCR.” Should be revised to 2.4 H. pylori detection by PCR

6. P5, lines 208~211, the unit symbol of the temperature is incorrect and requires modification.

7. In this manuscript, the hyphen was used to represent the scope such as “0.9-1.1” and “80-90%”, which is suggested to be revised to “0.9–1.1” and “80–90%”.

8. The references in this manuscript requires tighter editing. For example, “Helicobacter pylori” should be revised to be italic.

Author Response

                                                                                                                          Madrid, August 22th, 2022

Dear Reviewer,

 Please find enclosed our manuscript entitled “Evaluation of a new monoclonal chemiluminescent immunoassay stool antigen test for the diagnosis of Helicobacter pylori infection: A Spanish multicentre study”, which we would like you to consider for publication in the Special Issue of Journal of Clinical Medicine as an original paper.

After a careful revision of our article proposal, based on the suggestions made by the reviewers, we have proceeded to send it for a new evaluation. In the new manuscript, we have highlighted in red the modifications made to the original text.

We would like to express our sincere gratitude to the reviewers for their valuable work; their annotations have allowed us not only to significantly improve the manuscript but also to reflect on future research.

In the following of this letter, we detail how we have addressed the reviewers' suggestions in the new version of our article proposal. We hope that the work we have done will meet with the final approval of the Editorial Team. Should this not be the case, all authors are at your disposal to resolve any issues or to proceed with further revisions to the extent necessary.

  • 1) The first time the word/phrase is introduced, put the abbreviation in brackets after the word/phrase, such as “ROC” in Page 1, line 49. The full name of the “ROC” abbreviation should be given when it first appears.
  • Thank you very much for your comment, we have already modified the original text and added the full name.
  • The sentence has been changed from “Overall, 307 patients (145 naïve, 162 with confirmation of eradication) were analysed. Using recommended cut-off values (negative SAT <0.90, positive ≥1.10) the sensitivity, specificity, positive-predictive-value, negative-predictive-value and accuracy were 67%, 97%, 86%, 92% and 91%, respectively, obtaining an area under the ROC curve (AUC) of 0.85” to “Overall, 307 patients (145 naïve, 162 with confirmation of eradication) were analysed. Using recommended cut-off values (negative SAT <0.90, positive ≥1.10) the sensitivity, specificity, positive-predictive-value, negative-predictive-value and accuracy were 67%, 97%, 86%, 92% and 91%, respectively, obtaining an area under the receiver operating characteristic (ROC) curve (AUC) of 0.85.”
  • 2) The detection principle of the new monoclonal chemiluminescent immunoas-2 say stool antigen test method should be provided in this manuscript.

  • This has been described in material and methods in the sentence "Two hundred microlitres of the diluted sample (mixture of sample diluent and faeces) was incubated with paramagnetic particles coated with capture antibodies. Isoluminol-conjugated antibodies for pylori antigen were then added and incubated and the unbound material was washed away. The chemiluminescent flash reaction was then initiated and the chemiluminescent light was measured using a photomultiplier. Relative light units (RLU) were recorded. The RLU were proportional to the concentration of H. pylori faecal antigen present", but for clarification we have added this paragraph to the discussion "Luminescent transitions of exited molecules or atoms to a state of lower energy are characterised by electromagnetic radiation dissipated as photons in the ultraviolet (UV), visible or near-infrared region. These luminescent reactions are classified according to the energy source involved during the excitation step; thus, most classical light-emission reactions are referred to as bioluminescence (from in vivo systems), chemiluminescence (from a chemical reaction) as it is the case of this study protocol, electroluminescence (from electrochemical reaction) and photoluminescence (from UV, visible or near-infrared radiations). Chemiluminescence reactions are generally oxidoreduction processes and the exited compound, which is the reaction product, has a different chemical structure from the initial reactant".

  • 3) The detailed testing experimental steps of the new monoclonal chemiluminescent immunoas-2 say stool antigen test method should be supplemented in the section of “2. Materials and Methods”.

  • The complete protocol of the LIAISON® Meridian pylori SA Stool Antigen Test can be added as supplementary material in the material and methods section. Please find attached the document. We have included it as supplementary material (Supplementary Material Figure S1).

  • 4) This paper reports on the evaluation of a new monoclonal chemiluminescent immunoas-2 say stool antigen test for the diagnosis of Helicobacter pylori In a wider perspective, also other approaches, based on immobilization of pathogens or organelles on the sensor surface have been attempted to study the interactions of drugs with sensor components for the purpose of pathogen control or disease curing (Biosensors and Bioelectronics, 2022, 215: 114376; Sensors and Actuators B: Chemical, 2022, 355: 131284). These relevant literature references should be cited for the benefit of general readership.

  • Thank you very much for your comment. We have found these articles very interesting; we agree that they can complete and contribute with relevant information to our study. We have therefore decided to include and cite them in the discussion of the manuscript.

  • 5). P4, line 196, “H. pylori detection by PCR.” Should be revised to “2.4 H. pylori detection by PCR

  • Thank you very much, we have removed the full stop.

  • 6). P5, lines 208~211, the unit symbol of the temperature is incorrect and requires modification.

  • Thank you very much, we have corrected the temperature symbol from “ºC” to “°C.”

  • 7) In this manuscript, the hyphen was used to represent the scope such as “0.9-1.1” and “80-90%”, which is suggested to be revised to “0.9–1.1” and “80–90%

  • Reviewed and changed from “- “to “– “

  • 8). The references in this manuscript requires tighter editing. For example, “Helicobacter pylori” should be revised tobe italic.

  • Reviewed and changed to italics in all references.

Sincerely,

Elena Resina and Javier P. Gisbert

Round 2

Reviewer 2 Report

I have evaluated the REVISED VERSION of “Efficient production of high-molecular-weight hyaluronic acid with a two-stage fermentation”.  Authors have carefully evaluated the reviewers’ and editor’s comments and suggestions, responded to the suggestions point-by-point, and revised the manuscript accordingly.